# MCL-GAN: Generative Adversarial Networks with Multiple Specialized Discriminators

**Jinyoung Choi**[1,3]   **Bohyung Han**[1,2,3]
[1]ECE, [2]IPAI, [3]ASRI
Seoul National University, Korea
{jin0.choi,bhhan}@snu.ac.kr

## Abstract

We propose a framework of generative adversarial networks with multiple discriminators, which collaborate to represent a real dataset more effectively. Our approach facilitates learning a generator consistent with the underlying data distribution based on real images and thus mitigates the chronic mode collapse problem. From the inspiration of multiple choice learning, we guide each discriminator to have expertise in a subset of the entire data and allow the generator to find reasonable correspondences between the latent and real data spaces automatically without extra supervision for training examples. Despite the use of multiple discriminators, the backbone networks are shared across the discriminators and the increase in training cost is marginal. We demonstrate the effectiveness of our algorithm using multiple evaluation metrics in the standard datasets for diverse tasks.

## 1 Introduction

With recent advances in deep generative models, Generative Adversarial Networks (GANs) [1] and Variational Autoencoders (VAEs) [2] have shown impressive achievements in generation of high-dimensional realistic images as well as various computer vision applications including image-to-image translation [3–5], image inpainting [6], image super-resolution [7], *etc*. GANs have particularly received a lot of attention due to their interesting framework of minimax games, where two agents, a generator and a discriminator, compete against each other. Furthermore, compared to VAEs, GANs tend to produce acute, high-quality images. In theory, the generator learns the real data distribution by reaching an equilibrium point of the minimax game. However, in practice, the alternating training procedure does not guarantee the convergence to the optimal solution and often experiences mode collapsing, failing to cover the multiple modes of real data or, even worse, reaching trivial solutions.

This paper focuses on the mode collapse problem in training GANs by adopting multiple collaborating discriminators. Each discriminator learns to be specialized in a subset of reference data space, which is identified automatically during training, so the ensemble of discriminators provides not only the differentiation of fake data but also more accurate predictions over the clusters of real data. In this respect, a generator is encouraged to produce diverse modes that deceive a set of discriminators. To achieve these goals, we employ a Multiple Choice Learning (MCL) [8] framework to learn multiple discriminators and update the generator via a set of expert discriminators, where each discriminator is associated with a subset of the true and generated examples. Our approach, based on a single generator and multiple discriminators, is called MCL-GAN, which is optimized by the standard objective of GANs combined with the objective of MCL on the discriminator side.

There are several GAN literature that adopts multiple discriminators [9–14]. Among them, GMAN [10] is closely related to our approach in the sense that it utilizes an ensemble prediction of discriminators. It explores multi-discriminator extensions of GANs with diverse versions of the aggregated prediction of discriminators—from a harsh trainer to a lenient teacher with softened

criteria. Meanwhile, there exist significant differences in the method of ensembling from our approach. While GMAN focuses on the loss to the generator with parallel learning of discriminators, our strategy specializes in each discriminator for more informative feedback to the generator.

The training algorithm of the proposed approach is inspired by MCL [15], which is known to be effective in learning specialized models with high oracle accuracy in recognition tasks. Encouraged by this benefit, [16–19] apply MCL or its variations [20, 21] to the production of diverse and accurate outputs in several applications. For instance, Mun *et al*. [17] propose an MCL-KD framework to come up with a visual question answering (VQA) system based on multiple models that are specialized in different types of visual reasoning. DiverseNet [18] introduces a control parameter as an input that diversifies the outputs of the network with an MCL loss. While these works generate multiple outputs explicitly and select one of them or take their ensemble at inference time, our approach adopts a unique strategy for diversifying the mode with no additional cost at inference time.

The proposed method takes an advantage of MCL for GANs, which has hardly been explored before. Our approach does not require any supervision for model specialization, such as class labels and other conditions, unlike the aforementioned works. Our main contributions are summarized as follows:

- We propose a single-generator multi-discriminator GAN training algorithm to alleviate the mode collapse problem. Our approach provides simple yet effective learning objectives based on MCL to achieve the goal.

- We present a balanced discriminator assignment strategy to facilitate the robust convergence of models and preserve the multi-modality of training data, where the number of discriminators is determined adaptively.

- The proposed method is applicable to many GAN variants since there is no constraint on network architectures or loss functions. Our method requires small extra overhead and trains models efficiently via feature sharing in the discriminators.

## 2 Related work

This section discusses the mode collapse issue in GANs and the GAN models with multiple discriminators or generators.

### 2.1 Handling mode collapse for diversity

Many variations of GANs propose either novel metrics for the discriminator loss or better alternatives to the discriminator design. For example, LSGAN [22] substitutes the least square function for the binary cross-entropy loss as the discriminator objective. WGAN [23] introduces a critic function based on the Earth-Mover's distance rather than a binary classifier, and WGAN-GP [24] improves WGAN by adding a gradient penalty term. PacGAN [25] augments the discriminator's input by packing samples for a single label. EBGAN [26] models the discriminator as an energy function, which is optimized by the reconstruction loss as in autoencoders. BiGAN [27], ALI [28], VEEGAN [29], and Inclusive GAN [30] also learn reconstruction networks. In particular, VEEGAN [29] autoencodes the latent vectors to learn the inverse function of the generator, and maps both the true and generated data to the latent distribution, *e.g.*, a Gaussian distribution. Inclusive GAN [30] learns a generator by matching between real and fake examples in the feature space. The mode collapse and diversity issues of generated outputs have been addressed explicitly in [12, 31, 32]. They formulate the diversity metrics that encourage the mode exploration of the generators and derive the loss function using the metrics.

### 2.2 GAN with multiple generators

Another line of research is the integration of multiple generators [33–36]. This approach represents a data distribution with a mixture model enforcing each generator to cover a portion of the whole data space. It is naturally expected that mixture models approximate true distributions better than a single model, especially in high-dimensional spaces with multiple modes.

MAD-GAN [34] introduces an augmented classifier as a discriminator, which predicts whether a sample is real and which generator the sample is drawn from, to encourage individual generators to

learn distinctive modes. MGAN [35] has similar strategies to MAD-GAN but constructs a separate branch in its discriminator to perform the two tasks. MEGAN [36] adopts a gating network that produces a one-hot vector to select the generator creating the best example. P2GAN [37] sequentially adds a new generator to cover the missing modes of real data.

## 2.3 GAN with multiple discriminators

Multiple discriminators are often employed to improve the performance of a single generator [9–11, 13]. D2GAN [9] conducts a three-player minimax game, where two discriminators are trained for the completely opposite objectives, minimizing the Kullback-Leibler (KL) divergence and the inverse KL divergence between the true and generated data distributions. The balancing of the two losses plays a role in seeking desirable and diverse modes. Albuquerque *et al.* [13] propose a general multi-objective optimization framework in the scenario with multiple discriminators. They present a hypervolume maximization algorithm to obtain weighted gradients. Neyshabur *et al.* [14] train a GAN based on multiple projections. Each discriminator determines a random low-dimensional projection of a sample to address the instability of GAN training in high-dimensions. GMAN [10] presents diverse aggregation methods of multiple discriminators, where both hard and soft discriminator selection strategies are studied. Note that all the existing approaches learn the multiple discriminators independently. Thus, these discriminators may have strong correlations, which may not be appropriate for diversifying generated samples. Our approach, however, assigns each sample to the best-suited discriminator through interactions among the discriminators, and, consequently, each discriminator becomes an expert model for the assigned examples.

## 3 Multiple Choice Learning

We present the main idea of MCL [8] and its extensions briefly. Given a training dataset with $N$ samples, $\mathcal{D} = \{(\mathbf{x}_i, y_i)\}_{i=1}^N$, $M$ models, $\{f_m\}_{m=1}^M$, and a task-specific loss function, $\ell(\cdot, \cdot)$, MCL minimizes the following oracle loss:

$$\mathcal{L}_{\text{MCL}}(\mathcal{D}) = \sum_{i=1}^N \min_m \ell(y_i, f_m(\mathbf{x}_i)). \tag{1}$$

In other words, only the model with the smallest error out of $M$ candidates is selected for each example. This optimization process makes each model $f_m(\cdot)$ become an expert for a subset of $\mathcal{D}$, thus leads to forming a natural cluster in $\mathcal{D}$.

A weakness of MCL is the possible mistakes caused by overconfidence issues. If non-specialized models make wrong predictions with high confidence in the score aggregation process, the average scores are misleading and the ensemble model may result in poor-quality outputs. To alleviate this limitation, Confident Multiple Choice Learning (CMCL) [20] adopts a confident oracle loss that enforces the predictions of a non-specialized model to be uniformly distributed using KL divergence, denoted by $D_{\text{KL}}$. Assuming that $f_m(\cdot)$ predicts the output distribution given data point $\mathbf{x}$, *i.e.*, $P_m(y|\mathbf{x})$, the loss is modified as

$$\mathcal{L}_{\text{CMCL}}(\mathcal{D}) = \sum_{i=1}^N \sum_{m=1}^M v_{i,m} \ell(y_i, P_m(y|\mathbf{x}_i)) + \beta(1 - v_{i,m}) D_{\text{KL}}(\mathcal{U}(y) \| P_m(y|\mathbf{x}_i)), \tag{2}$$

where $\mathcal{U}(y)$ is the uniform distribution and the flag variable $v_{i,m} \in \{0, 1\}$ allows the choices of the specialized models. Note that, if $\sum_{m=1}^M v_{i,m} = k$ ($k < M$), each example is assigned to $k$ models.

## 4 MCL-GAN

We describe our GAN extension with a generator $G(\cdot; \theta)$ and $M$ discriminators $\{D_m(\cdot; \phi_m)\}_{m=1}^M$. Let $p_z$ and $p_d$ be the distributions in latent space and real data space, respectively. Given $\mathbf{z} \sim p_z$, the generator produces a sample $\tilde{\mathbf{x}} = G(\mathbf{z}; \theta)$ and each of the $M$ discriminators makes a prediction on a real example $\mathbf{x} \sim p_d$ or the fake sample $\tilde{\mathbf{x}}$. Each prediction, $D_m(\mathbf{x}; \phi_m)$, ranges in $[0, 1]$ and represents the probability that $\mathbf{x}$ belongs to the true data distribution. We now describe the loss functions $\mathcal{L}^D$ and $\mathcal{L}^G$ for training discriminators and the generator, respectively.

## 4.1 Expert training

Assuming that we draw $N_d$ real data and generate $N_g$ examples in each training batch, denoted by $\mathbf{x}$ and $\tilde{\mathbf{x}}$, respectively, each network is trained as follows.

**Discriminators**   Expert discriminators are the ones that predict the highest scores for each example. With the indicator variable $v_{i,m}$ for a real example $\mathbf{x}_i$, the discriminators are trained to minimize the following loss function:

$$\mathcal{L}_{\mathrm{e}}^D(\mathbf{x}) = -\sum_{i=1}^{N_d} \sum_{m=1}^{M} v_{i,m} \log(D_m(\mathbf{x}_i; \phi_m)), \tag{3}$$

where we choose $k$ experts out of $M$ discriminators for each example, *i.e.*, $\sum_{m=1}^{M} v_{i,m} = k$. For a fake sample, since all discriminators have to identify it correctly, the following loss is added to (3):

$$\mathcal{L}_{\mathrm{e}}^D(\tilde{\mathbf{x}}) = -\sum_{j=1}^{N_g} \sum_{m=1}^{M} \log(1 - D_m(G(\mathbf{z}_j); \phi_m)). \tag{4}$$

**Generator**   We train the generator using the gradients received from the expert models to encourage the generator to find the closest mode given $\mathbf{z}$. With another indicator variable $u_{j,m}$ for $\mathbf{z}_j$, the expert loss for the generator is given by

$$\mathcal{L}_{\mathrm{e}}^G(\tilde{\mathbf{x}}) = \sum_{j=1}^{N_g} \sum_{m=1}^{M} u_{j,m} \log(1 - D_m(G(\mathbf{z}_j; \theta)); \phi_m) \quad \text{and} \quad \sum_{m=1}^{M} u_{j,m} = k. \tag{5}$$

## 4.2 Non-expert training

The non-expert discriminators should not be overconfident to real examples while it is desirable to produce higher scores for real samples than fake ones. For this requirement, we give a uniform soft label, *e.g.*, $y = [0.5, 0.5]$ for non-expert discriminators and regularize them with a certain weight. To be precise, we obtain the following non-expert loss term corresponding to (3):

$$\mathcal{L}_{\mathrm{ne}}^D(\mathbf{x}) = \sum_{i=1}^{N_d} \sum_{m=1}^{M} (1 - v_{i,m}) \ell_{\mathrm{ce}}(D_m(\mathbf{x}_i), y), \tag{6}$$

where $v_{i,m}$ is the same indicator variable defined in (3) and $\ell_{\mathrm{ce}}(\cdot, y)$ is the cross-entropy loss function given a target label $y$. The other counterpart for (5) is derived similarly as

$$\mathcal{L}_{\mathrm{ne}}^G(\tilde{\mathbf{x}}) = \sum_{j=1}^{N_g} \sum_{m=1}^{M} (1 - u_{j,m}) \ell_{\mathrm{ce}}(D_m(G(\mathbf{z}_j)), y). \tag{7}$$

The non-expert model training is effective to handle the overconfidence issue, but the model may still suffer from the data deficiency problem frequently happening in the standard MCL algorithms because each discriminator can observe only a subset of the whole dataset. To alleviate this limitation, our discriminators share the parameters of all layers in the feature extractor while branching the last layer only. This implementation is also sensible since the discriminators partially have the same objective to distinguish fake examples. The common representations of all real samples are prone to be learned in the earlier layers despite being clustered in different subsets, whereas the critical information for high-level understanding is often encoded in deeper layers. Moreover, the number of model parameters and training time are saved significantly while still taking advantage of ensemble learning.

## 4.3 Balanced assignment of discriminators

On top of the adversarial losses, we introduce another loss for balanced updates of discriminators. As there is no supervision about the specialization target of a discriminator, *e.g.*, class labels or feature embeddings, it may be difficult to reasonably distribute real samples to expert models from

the beginning. Since the abilities of individual discriminators are severely off-balanced, they are highly prone to assign all samples to a few specific models. Especially at an early phase of training, the model's capability is more sensitive to the number of updates in the discriminators.

To tackle this challenge, we propose another loss, called the balance loss, which gives discriminators equal chances for updates. Let the selection of expert discriminators approximately follow a categorical distribution with a parameter $\boldsymbol{\mu} = [\mu_1, \ldots, \mu_M]$. Then the loss is computed by the KL divergence of the probability distribution of discriminators for being selected as experts from $\boldsymbol{\mu}$. To obtain the probability for discriminator selection, we apply the $\mathtt{softmax}$ function to the vector of $M$ predictions from discriminators for each example since the discriminator with the highest score is guaranteed to be chosen as an expert. We average these probability vectors over the training batch, $i.e.$, $\mathbf{q} = \frac{1}{N_d} \sum_{i=1}^{N_d} \mathbf{s}([D_1(\mathbf{x}_i), \ldots, D_M(\mathbf{x}_i)]; \tau)$, where $\mathbf{s}(\cdot; \tau)$ denotes a vector-valued $\mathtt{softmax}$ function with a temperature $\tau$ given a logit. To sum up, the balance loss is given by

$$\mathcal{L}_{\mathrm{bal}}^D(\mathbf{x}) = D_{\mathrm{KL}}(\boldsymbol{\mu} \| \mathbf{q}). \tag{8}$$

In practice, we set $\mu_m = \frac{1}{M}, \forall m$ to update the discriminators evenly, which is because the true distribution is unavailable. This assumption may not be congruent to the real distribution of the dataset and excessively forced assignment would not result in an optimal clustering for specialization. We, therefore, decrease the weight for the balance loss gradually during training. Eventually, each example will be naturally assigned to its best model with a very small weight of the balance loss. This adjustment helps stabilize training and naturally cluster the reference data. Note that the models are balanced within a few epochs and the weight reduction helps generate higher quality samples.

Likewise, a small constraint on the distribution of the generator's output facilitates balanced generation while preventing the statistics of generated samples from being skewed. For this case, we use the distribution of the discriminators' assignments $\mathbf{q}$ instead of arbitrarily chosen $\boldsymbol{\mu}$, $i.e.$,

$$\mathcal{L}_{\mathrm{bal}}^G(\tilde{\mathbf{x}}) = D_{\mathrm{KL}}(\mathbf{q} \| \mathbf{o}). \tag{9}$$

where $\mathbf{o} = \frac{1}{N_g} \sum_{j=1}^{N_g} \mathbf{s}([D_1(G(\mathbf{z}_j)), \ldots, D_M(G(\mathbf{z}_j))]; \tau)$.

### 4.4 Total loss

Altogether, the total losses for discriminators and the generator are summarized respectively as

$$\mathcal{L}^D = \mathcal{L}_{\mathrm{e}}^D + \alpha \mathcal{L}_{\mathrm{ne}}^D + \beta_d \mathcal{L}_{\mathrm{bal}}^D \quad \text{and} \quad \mathcal{L}^G = \mathcal{L}_{\mathrm{e}}^G + \alpha \mathcal{L}_{\mathrm{ne}}^G + \beta_g \mathcal{L}_{\mathrm{bal}}^G, \tag{10}$$

where $\alpha$ and $\{\beta_d, \beta_g\}$ are hyperparameters. Note that the proposed approach is applicable to any GAN formulations with the corresponding adversarial losses.

### 4.5 Choice of number of discriminators

A remaining concern is how to find the optimal number of discriminators while such information is not available in general as in many clustering tasks. If the number of discriminators is much larger than the optimal one, it is more desirable to focus on training a subset of discriminators than dividing the dataset into many minor clusters forcefully.

To ease this issue, we augment the $\ell_1$ regularization loss to the discriminator side with a weight $\gamma$ and encourage the sparsity in the discriminator selection for more desirable clustering results. Hence, even in the case that we are given an excessively large number of discriminators, our algorithm converges at good points by using a small subset of discriminators in practice. It is true that this strategy may not always lead to the optimal number of discriminators and have conflicts with the balance loss in (8). However, the balance loss fades away as training goes on, and our model identifies a proper number of clusters by deactivating a fraction of discriminators adaptively. This sparsity loss would be useful when we learn on real world datasets, where the examples are drawn from unknown distributions.

## 5 Experiments

We evaluate the performance of MCL-GAN on unconditional and conditional image generation. In the results reported throughout this section, the asterisk (*) denotes the copy from other works.

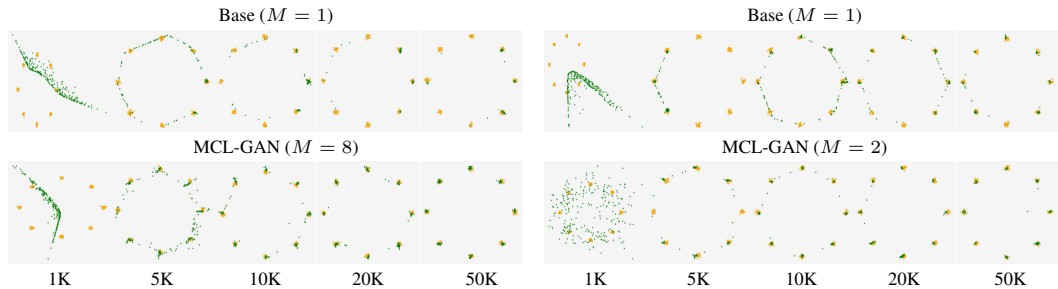

Figure 1: Snapshots of 256 random samples drawn from the generators of the baseline and MCL-GAN with (left) the standard GAN loss and (right) the Hinge loss after 1K, 5K, 10K, 20K and 50K steps. Data sampled from the true distribution are in orange while the generated ones are in green.

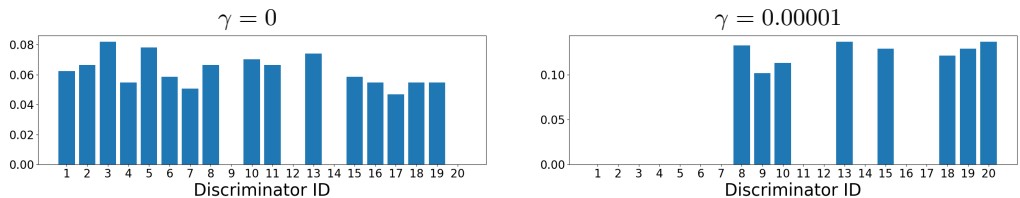

Figure 2: Effect of the $\ell_1$ loss weight in MCL-GAN ($\gamma$). The graphs show the ratio of training examples associated with each discriminator.

## 5.1 Results on synthetic datasets

We first perform toy experiments to verify the main idea of MCL-GAN intuitively. We consider a mixture of 8 isotropic Gaussians in 2D whose centers are located on the circumference of a circle with a radius $\sqrt{2}$ while their standard deviation in each dimension is 0.05. We employ 8 discriminators for training with the standard GAN loss while utilizing 2 discriminators with the Hinge loss [38]. We choose one expert discriminator for each sample ($k = 1$) in all experiments.

Figure 1 illustrates the snapshots of generated examples, given by the baseline methods and MCL-GANs, through iterations. While the base models ($M = 1$) fail to cover all 8 modes, MCL-GANs manage to identify diverse modes quickly and produce samples belonging to all modes eventually. The models trained with the Hinge loss tend to generate more diverse samples even only with a single discriminator; MCL-GAN also requires less discriminators in the Hinge loss case to reconstruct the original data distribution. Figure 2 illustrates how adaptively MCL-GANs select discriminators, given an excessive number of discriminators, *e.g.*, $M = 20$. They utilize 8 and 16 expert discriminators when $\gamma = 0$ and $10^{-5}$, respectively; each mode tends to be associated with 1 or 2 discriminators depending on the value of $\gamma$. We visualize the detailed mapping between examples and discriminators in Appendix A.1. This observation implies that MCL-GAN covers all modes effectively while the $\ell_1$ loss helps identify the proper number of discriminators to generate high-fidelity data.

## 5.2 Unconditional GANs on image dataset

### 5.2.1 Experiment setup and evaluation protocol

We run the unconditional GAN experiment on four distinct datasets including MNIST [39], Fashion-MNIST [40], CIFAR-10 [41] and CelebA [42], where two types of network architectures, DC-GAN [43] and StyleGAN2 [44], are employed. Images are resized to 32×32 except for the CelebA dataset, for which DCGAN and StyleGAN2 adopt 64×64 and 128×128 images, respectively. For the StyleGAN2 experiments on CelebA, we use the first and last 30K images from the "align&cropped" version of the train and validation splits following [30]. We apply our method to the DCGAN architecture with three different GAN loss functions: the vanilla GAN loss [1], the LSGAN loss [22], and the Hinge loss [38]. Appendix C describes more details of our setting.

We adopt Precision Recall Distribution (PRD) [46] and Frèchet Inception Distance (FID) [47] as our evaluation metrics. The PRD curve provides a more credible assessment of generative models than

Table 1: Precision and recall scores from PRD curves on MNIST, Fashion-MNIST and CelebA datasets with the DCGAN architecture. Note that GMAN fails to converge with the Hinge loss.

| Method | Loss | $M$ | MNIST | | Fashion-MNIST | | CelebA | | |
| --- | --- | --- | --- | --- | --- | --- | --- | --- | --- |
| | | | Rec.↑ | Prec.↑ | Rec.↑ | Prec.↑ | FID ↓ | Rec.↑ | Prec.↑ |
| Base (DCGAN) [43] | | 1 | 0.896 | 0.778 | 0.936 | 0.900 | 30.93 | 0.834 | 0.839 |
| GMAN [10] | | 5 | 0.968 | 0.976 | 0.909 | **0.955** | 31.66 | 0.888 | 0.873 |
| GMAN [10] | GAN | 10 | 0.964 | **0.977** | 0.928 | 0.946 | 22.45 | 0.921 | 0.923 |
| MCL-GAN | | 5 | **0.985** | 0.977 | **0.977** | 0.929 | **16.88** | 0.955 | 0.957 |
| MCL-GAN | | 10 | 0.976 | 0.975 | 0.964 | 0.914 | 21.18 | 0.940 | 0.938 |
| Base (DCGAN) [43] | | 1 | 0.977 | 0.957 | 0.928 | 0.866 | 19.87 | 0.923 | 0.943 |
| GMAN [10] | LSGAN [22] | 10 | 0.966 | 0.973 | 0.953 | **0.952** | 22.72 | 0.934 | 0.906 |
| MCL-GAN | | 10 | **0.983** | **0.980** | **0.963** | 0.911 | **17.81** | 0.950 | 0.952 |
| Base (DCGAN) [43] | | 1 | 0.790 | 0.785 | 0.936 | 0.853 | 23.56 | 0.905 | 0.883 |
| MCL-GAN | Hinge [38] | 5 | 0.957 | 0.965 | **0.959** | **0.916** | **20.49** | 0.914 | 0.925 |
| MCL-GAN | | 10 | **0.978** | **0.968** | 0.949 | 0.885 | 21.23 | **0.928** | **0.931** |

Table 2: FID scores on CIFAR-10 with the DCGAN architecture, where † denotes the usage of the different DCGAN architecture adopted in [45].

| Method | # Disc. | # Gen. | FID ↓ | Requirements |
| --- | --- | --- | --- | --- |
| Base (DCGAN)* [43] | 1 | 1 | 37.7 | |
| GMAN [10] | 10 | 1 | 37.11 | |
| Albuquerque *et al.* [13] | 10 | 1 | 30.26 | |
| MCL-GAN | 10 | 1 | **26.87** | |
| MGAN* [35] | 1 | 10 | 26.7 | Multiple generators for inference |
| MSGAN* [12] | 1 | 1 | 28.73 | Class labels for training |
| Base (GAN)*† | 1 | 1 | 31.90 | |
| Self-conditioned GAN*† [45] | 1 | 1 | 18.70 | Clustering (with 100 clusters) |
| MCL-GAN† | 5 | 1 | **17.15** | |

Table 3: KL divergence and LPIPS using pretrained classifiers.

| Method | $M$ | MNIST | | Fashion-MNIST | | CIFAR-10 | |
| --- | --- | --- | --- | --- | --- | --- | --- |
| | | KL ↓ | LPIPS↑ | KL ↓ | LPIPS ↑ | KL ↓ | LPIPS ↑ |
| Base (DCGAN) [43] | 1 | 0.0268 | 0.0257 | 0.0437 | 0.0233 | 0.0532 | 0.0562 |
| GMAN [10] | 5 | **0.0072** | 0.0238 | 0.0587 | 0.0289 | 0.1084 | **0.0594** |
| MCL-GAN | 5 | 0.0127 | **0.0259** | **0.0194** | **0.0294** | **0.0474** | 0.0562 |

single-valued metrics since it quantifies the model in two folds—mode coverage (recall) and quality of generated samples (precision). We measure the recall and precision of each model by computing the $F_8$ and $F_{1/8}$ scores from the PRD curve, respectively. We prefer higher scores and they are equal to 1 when the generated data distribution is identical to the reference. FID is a popular evaluation metric of generative models, based on the distance between two datasets with multivariate Gaussian assumption. Lower FID scores mean that generated samples are closer to the reference dataset.

### 5.2.2 DCGAN backbone

Table 1 summarizes the precision and recall scores of our methods compared to other models with three different GAN objectives, when the number of discriminators is set to 5 or 10 ($M = 5$ or 10) and the number of experts is 1 ($k = 1$). MCL-GAN achieves outstanding performance in terms of both metrics compared to the baseline and GMAN, which is an existing approach adopting multiple discriminators, on MNIST and CelebA. For Fashion-MNIST, we observe that MCL-GAN focuses on the mode coverage (diversity) while GMAN cares about image quality rather than diversity.

As presented in Table 2, MCL-GAN outperforms the compared GAN models (based on DCGAN) by large margins in terms of FID scores on CIFAR-10, while being competitive with MGAN, which relies on multiple generators. MCL-GAN also achieves better results than a clustering-based approach, *e.g.*, self-conditioned GAN [45], which requires additional computation due to clustering during training and the extra model parameters for conditioning the generator's inputs with cluster membership. The results imply that MCL-GAN is effective to maintain multi-modality in the underlying distribution with a relatively small memory footprint and without extra supervision.

To analyze the semantic quality of generated images, we present their classification results given by the pretrained classifiers in Table 3. We measure how much the predicted label distribution in each

Table 4: FID, precision and recall scores on CIFAR-10 and CelebA datasets with the StyleGAN2 architecture, where 10 and 5 discriminators are adopted, respectively, while $k = 1$.

| Method | CIFAR-10 | | | CelebA30K* | | | | | |
| | FID↓ | Rec.↑ | Prec.↑ | FID↓ | | Rec.↑ | | Prec.↑ | |
| | - | - | - | Train | Val | Train | Val | Train | Val |
| Base (StyleGAN2) [44] | 9.06 | 0.979 | 0.984 | 9.37 | 9.49 | 0.730 | 0.741 | 0.855 | 0.844 |
| Inclusive GAN [30] | - | - | - | 11.56 | 11.28 | 0.849 | 0.848 | 0.927 | 0.941 |
| MCL-GAN | **7.13** | **0.985** | **0.989** | **8.41** | **8.61** | **0.988** | **0.990** | **0.985** | **0.983** |

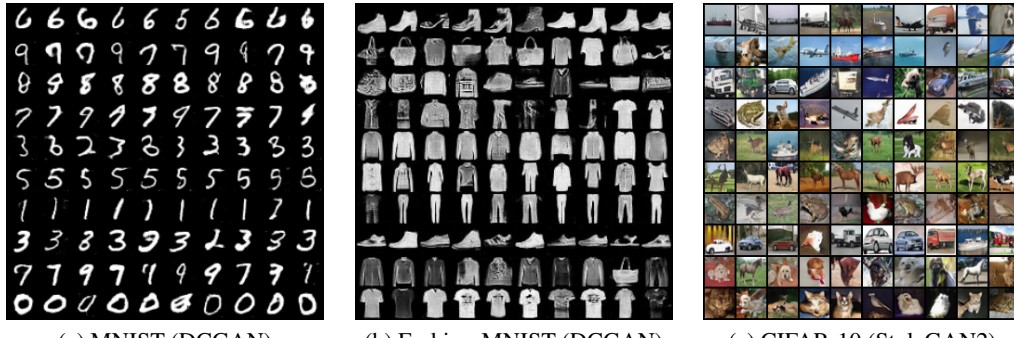

| (a) MNIST (DCGAN) | (b) Fashion-MNIST (DCGAN) | (c) CIFAR-10 (StyleGAN2) |

Figure 3: Generated image clusters by MCL-GAN. Each row represents the cluster associated with each discriminator ($M = 10$). Note that the images in the same row often have similar shapes and semantics but are not necessarily in the same class.

tested dataset deviates from the true (uniform) distribution using the KL divergence. We also calculate the average of class-wise LPIPS to measure the intra-class diversity of generated images. The overall results are favorable to MCL-GAN but there exist some misleading points. For example, GMAN tends to achieve overly high class-wise LPIPS scores due to many out-of-distribution examples that confuse the classifier, as illustrated in Appendix B.1.

### 5.2.3 StyleGAN2 backbone

Table 4 presents that MCL-GAN is also effective in the state-of-the-art backbone model and outperforms StyleGAN2 [44] and its variation, Inclusive GAN [30], in terms of all metrics. Inclusive GAN employs the sample-wise reconstruction loss by regarding each image as a mode. This strategy appears to improve recall but the model may suffer from sampling bias and scalability issues by estimating overly complex distributions. The results imply that MCL-GAN enhances the convergence and the reproducibility of large GAN models, practically leading to improved performance.

### 5.2.4 Discriminators specialization

**Qualitative results**   Figure 3 qualitatively presents how successfully the discriminators in MCL-GAN are specialized to subsets of datasets. We learn the model with 10 discriminators, and illustrate the generated images with their memberships to discriminators; the images in the same row belong to the same discriminator. We observe the semantic consistency of images within the same row in both MNIST and Fashion-MNIST clearly.

**Analysis using attribute annotations**   To examine how effectively the discriminators represent the real data, we analyze the discriminator specialization statistics of MCL-GAN using the 40 binary attributes of each image in the CelebA dataset. We first compute the normalized histogram of discriminator assignments for the images with a certain attribute, and then compute the cosine similarity between the histograms to obtain a $40 \times 40$ dimensional similarity matrix, where 40 is the number of attributes. Figure 4 illustrates the similarity matrix, where we sample a subset of attributes for better visualization. We discover that the discriminator specialization statistics highly coincide with the latent semantic similarities of examples. Specifically, the four attributes more relevant to female—*Arched_Eyebrows, Wearing_Earings, Wearing_Lipstick, and Heavy_Makeup*—are clustered together, while another set of four attributes such as *Goatee, Male, Mustache,* and *Sideburns* construct

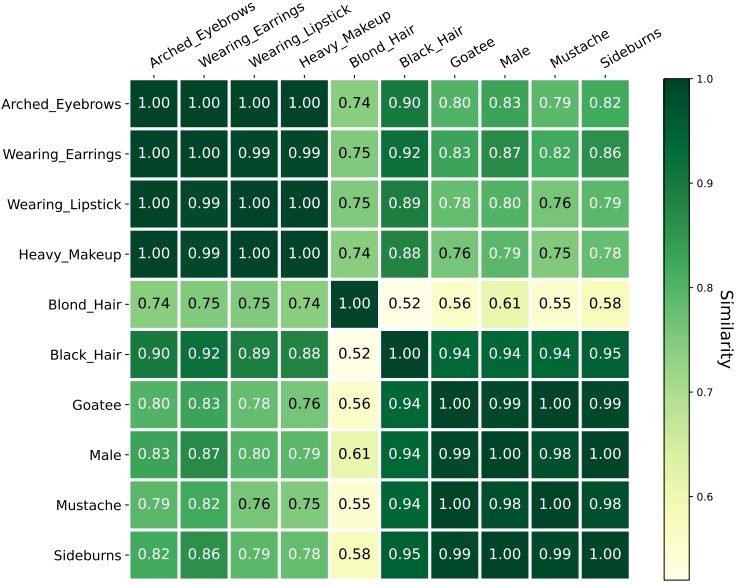

Figure 4: The cosine similarity matrix among the attribute representations obtained by the discriminator assignments of MCL-GAN on the CelebA dataset.

Table 5: Quantitative results of conditional GANs in the image-to-image translation tasks.

| Dataset | Metric | DRIT* | +MS (DRIT++)* [12] | +MCL | +MCL+MS |
|---|---|---|---|---|---|
| Summer $\rightarrow$ Winter | FID $\downarrow$ | $57.24 \pm 2.03$ | $51.85 \pm 1.16$ | $53.77 \pm 1.36$ | $\mathbf{49.74 \pm 2.74}$ |
| | NDB $\downarrow$ | $25.60 \pm 1.14$ | $\mathbf{22.80 \pm 2.96}$ | $25.40 \pm 1.14$ | $30.00 \pm 2.55$ |
| | JSD $\downarrow$ | $0.066 \pm 0.005$ | $0.046 \pm 0.006$ | $\mathbf{0.036 \pm 0.004}$ | $0.044 \pm 0.005$ |
| | LPIPS $\uparrow$ | $0.115 \pm 0.000$ | $0.147 \pm 0.001$ | $0.199 \pm 0.002$ | $\mathbf{0.263 \pm 0.003}$ |
| Winter $\rightarrow$ Summer | FID $\downarrow$ | $47.37 \pm 3.25$ | $46.23 \pm 2.45$ | $49.41 \pm 1.29$ | $\mathbf{41.94 \pm 1.43}$ |
| | NDB $\downarrow$ | $30.60 \pm 2.97$ | $27.80 \pm 3.03$ | $\mathbf{23.40 \pm 1.52}$ | $24.20 \pm 3.27$ |
| | JSD $\downarrow$ | $0.049 \pm 0.009$ | $0.038 \pm 0.004$ | $0.033 \pm 0.002$ | $\mathbf{0.030 \pm 0.005}$ |
| | LPIPS $\uparrow$ | $0.097 \pm 0.000$ | $0.118 \pm 0.001$ | $0.153 \pm 0.001$ | $\mathbf{0.248 \pm 0.001}$ |
| Cat $\rightarrow$ Dog | FID $\downarrow$ | $22.74 \pm 0.28$ | $16.02 \pm 0.30$ | $20.64 \pm 0.13$ | $\mathbf{15.36 \pm 0.16}$ |
| | NDB $\downarrow$ | $42.00 \pm 2.12$ | $27.20 \pm 0.84$ | $29.80 \pm 1.10$ | $\mathbf{22.20 \pm 2.77}$ |
| | JSD $\downarrow$ | $0.127 \pm 0.003$ | $0.084 \pm 0.002$ | $0.048 \pm 0.002$ | $\mathbf{0.031 \pm 0.002}$ |
| | LPIPS $\uparrow$ | $0.245 \pm 0.002$ | $0.280 \pm 0.002$ | $0.511 \pm 0.000$ | $\mathbf{0.553 \pm 0.000}$ |
| Dog $\rightarrow$ Cat | FID $\downarrow$ | $62.85 \pm 0.21$ | $29.57 \pm 0.23$ | $\mathbf{20.61 \pm 0.05}$ | $27.16 \pm 0.20$ |
| | NDB $\downarrow$ | $41.00 \pm 0.71$ | $31.00 \pm 0.71$ | $\mathbf{16.40 \pm 0.89}$ | $20.20 \pm 1.48$ |
| | JSD $\downarrow$ | $0.272 \pm 0.002$ | $0.068 \pm 0.001$ | $\mathbf{0.024 \pm 0.001}$ | $0.031 \pm 0.001$ |
| | LPIPS $\uparrow$ | $0.102 \pm 0.001$ | $0.214 \pm 0.001$ | $0.429 \pm 0.001$ | $\mathbf{0.482 \pm 0.000}$ |

Table 6: Quantitative results of conditional GANs in the text-to-image synthesis task.

| Dataset | Metric | StackGAN++* | +MS* [12] | +MCL | +MCL+MS |
|---|---|---|---|---|---|
| CUB-200-2011 | FID $\downarrow$ | $25.99 \pm 4.26$ | $25.53 \pm 1.83$ | $\mathbf{22.91 \pm 0.80}$ | $25.44 \pm 0.41$ |
| | NDB $\downarrow$ | $38.20 \pm 2.39$ | $30.60 \pm 2.51$ | $28.80 \pm 3.63$ | $\mathbf{23.20 \pm 3.03}$ |
| | JSD $\downarrow$ | $0.09 \pm 0.01$ | $0.07 \pm 0.00$ | $0.08 \pm 0.00$ | $\mathbf{0.05 \pm 0.00}$ |
| | LPIPS $\uparrow$ | $0.36 \pm 0.00$ | $0.37 \pm 0.01$ | $\mathbf{0.63 \pm 0.00}$ | $0.62 \pm 0.00$ |

a strong group. These results support that the individual discriminators learned by MCL-GAN are specialized to reasonable subsets of the dataset, which is helpful to generate more realistic images.

## 5.3 Conditional image synthesis

We apply MCL-GAN to image-to-image translation and text-to-image synthesis tasks, which require complex architectures to generate high-resolution images. In this experiment, we adopt MSGAN [12], a technique with a mode-seeking regularizer, as an additional component for alleviating the mode collapse in conditional GANs. Then, we observe whether the mode seeking and our multiple choice learning create synergy, based on FID, NDB/JSD [48], and LPIPS [49] following [12]. Note that NDB counts the number of statistically different bins based on the clusters made by $k$-means clustering.

Table 7: Comparison with other discriminator assignment strategies such as Minimum, Random, and GT-Assign. GT-Assign links an expert discriminator with real samples using the ground-truth class labels under our multi-discriminator framework but is unrealistic due to the requirement of the labels.

| Strategy | $M$ | $k$ | MNIST | | Fashion-MNIST | | CelebA | |
| --- | --- | --- | --- | --- | --- | --- | --- | --- |
| | | | Rec.↑ | Prec.↑ | Rec.↑ | Prec.↑ | Rec.↑ | Prec.↑ |
| Base (DCGAN) [43] | 1 | 1 | 0.896 | 0.778 | 0.936 | 0.900 | 0.834 | 0.839 |
| Minimum | 5 | 1 | 0.913 | 0.904 | 0.943 | 0.906 | 0.945 | 0.893 |
| Random | 5 | 1 | 0.971 | 0.954 | 0.930 | 0.917 | 0.930 | 0.946 |
| MCL-GAN | 5 | 1 | **0.985** | **0.977** | **0.977** | 0.929 | **0.955** | **0.957** |
| GT-Assign | 10 | 1 | 0.978 | 0.966 | 0.969 | **0.935** | - | - |

**Image-to-image translation** We choose DRIT [4, 5], an unpaired image-to-image translation method based on the cycle consistency, as our baseline. We employ MCL-GAN with $M = 3$ and $k = 1$ for distinguishing real and translated images. As shown in Table 5, MCL-GAN significantly improves the diversity measure, LPIPS, while achieving high-fidelity data generation performance in terms of other metrics. In particular, our approach works better on a more challenging task, cat $\rightleftharpoons$ dog, since it effectively handles the changes in both object shape and texture across domains by specializing discriminators to a subset of modes. Note that the integration of MCL is mostly beneficial and the addition of the mode-seeking module often leads to extra performance gains.

**Text-to-image synthesis** This experiment is based on StackGAN++ [50] trained on CUB-200-2011 [51] with a mode-seeking regularizer. StackGAN++ has a hierarchical structure with a specialized pair of a discriminator and a generator to a certain image resolution. We adopt its 3-stage version and trains an MCL-GAN with $M = 3$ and $k = 1$ only at the last stage, which handles images with size $256 \times 256$. Table 6 illustrates that the integration of MCL improves performance consistently, especially in terms of the diversity measure, LPIPS, where we observe the merit of the mode-seeking regularizer via the combination with MCL.

## 5.4 Discriminator assignment strategies

MCL-GAN is a model-agnostic ensemble algorithm with multiple discriminators, which is conceptually advantageous to other discriminator assignment strategies. In practice, our specialized discriminators to the subsets of training data outperforms independent training of discriminators on the whole dataset (as in GMAN) or different discriminator assignment strategies such as minimum-score discriminator selections, opposite to MCL-GAN, and random selections. Also, the performance of MCL-GAN is as competitive as the method assigning discriminators based on the ground-truth class labels, which exhibits the reliability of the discriminator specialization by MCL. Table 7 presents that MCL-GAN achieves outstanding performance compared to other strategies especially for the recall metric, even compared with GT-Assign without relying on class labels for discriminator specialization. The proposed approach is also efficient since it is free from any time-consuming clustering procedure for sample assignments to discriminators during training, and incurs marginal extra cost despite the use of multiple discriminators because the discriminators share the feature extractor.

## 6 Conclusion

We presented a generative adversarial network framework with multiple discriminators, where each discriminator behaves as an expert classifier and covers a separate mode in the underlying distribution. This idea is implemented by incorporating the concept of multiple choice learning. The combination of generative adversarial network and multiple choice learning turns out to be effective to alleviate the mode collapse problem. Also, the integration of the sparsity loss encourages our model to identify the proper number of discriminators and estimate a desirable distribution with low complexity. We demonstrated the effectiveness of the proposed algorithm on various GAN models and datasets.

**Acknowledgements** This work was partly supported by Samsung Electronics Co., Ltd., and the Institute of Information & communications Technology Planning & Evaluation (IITP) grants [No.2022-0-00959, (Part 2) Few-Shot Learning of Causal Inference in Vision and Language for Decision Making; No.2021-0-01343, Artificial Intelligence Graduate School Program (Seoul National University)] and the National Research Foundation of Korea (NRF) grant [No.2022R1A5A708390811, Trustworthy Artificial Intelligence] funded by the Korean government (MSIT).

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
