# OpenReview forum: "MCL-GAN: Generative Adversarial Networks with Multiple Specialized Discriminators"
_NeurIPS.cc/2022/Conference — NeurIPS 2022 Accept_

### Official Review · Reviewer_Hs6x · 2022-07-06

**Rating:** 6
**Confidence:** 3
**Soundness:** 3 good
**Presentation:** 3 good
**Contribution:** 3 good

**Summary:**

The authors propose a multi-discriminator generative adversarial network to address the issue of mode collapse. The discriminators are tied together through parameter sharing which helps reduce the training time.


**Questions:**

My major concern is whether the proposed method is overfitting on the training set due to several specialized discriminators. Please see and address point 1 in weakness above.

**Limitations:**

The authors have discussed the limitations of their work and potential negative societal impact. According to the authors:
"we believe that the proposed approach can alleviate the bias and fairness issues by identifying the minority groups in a dataset effectively."

How does the proposed method identify the minority groups in a dataset is not very clear to me.

**Strengths And Weaknesses:**

Strength:

1.	Aim to solve the problem of mode collapse which is of interest to the community.

2.	The number of discriminators is determined adaptively.

3.	The proposed model is generic and can be applied to various variants of GANS as there is no architecture or loss specific constraints.

Weakness:

1.	Since the proposed model leverages multiple specialized discriminators, I suspect that the generator might overfit the training data. I request the authors to provide visualization of nearest neighbours (from the training split) of some randomly generated samples. They should also compute and report (1) Pixel Memorization Score (PMS) and (2) Inception Memorization Score (IMS) as proposed in the following paper.
https://proceedings.mlr.press/v161/mondal21a.html

2.	Although the authors report various metrics, there has been several drawbacks identified with the precision/recall metric. I suggest the authors to compute and compare density/coverage score instead as proposed in:
http://proceedings.mlr.press/v119/naeem20a/naeem20a.pdf

---

> ### Author Response · Authors · 2022-08-02
> **Response #1**
>
> We appreciate the positive feedback and helpful suggestions. Below are the additional evaluation results that address the reviewer’s concerns.
>
> 1. We compute PMS and IMS following [1], and report them in the Table below. If the generator is overfitted to the training data, the scores between generated and train data (Gen-Tr) would be less than those between generated and test data (Gen-Te). The results imply that MCL-GAN does not memorize the training data for the high-fidelity generation. We also provide visualization of the nearest neighbors of some generated samples by MCL-GAN trained on 30k CelebA images with StyleGAN2 backbone in the supplementary file (please see Figure C of figures.pdf).
> |            |               |           | Tr-Tr |       | Gen-Tr |       | Tr-Te |       | Gen-Te |       |
> |:-----------|:--------------|:----------|:-----:|:-----:|:------:|:-----:|:-----:|:-----:|:------:|:-----:|
> |**Backbone**|**Dataset**    |**Method** |**PMS**|**IMS**|**PMS** |**IMS**|**PMS**|**IMS**| **PMS**|**IMS**|
> | DCGAN      | MNIST         | MCL-GAN   |  4.50 |  7.03 |   4.95 |  8.10 |  4.56 |  7.11 |   4.95 |  8.08 |
> |            |               | GMAN      |       |       |   5.25 |  8.25 |       |       |   5.25 |  8.24 |
> | DCGAN      | Fashion-MNIST | MCL-GAN   |  4.03 |  9.20 |   4.14 |  9.58 |  4.03 |  9.21 |   4.14 |  9.57 |
> |            |               | GMAN      |       |       |   3.96 |  9.58 |       |       |   3.97 |  9.58 |
> | StyleGAN2  | CIFAR-10      | MCL-GAN   |  9.79 | 12.51 |   9.76 | 12.71 |  9.77 | 12.51 |   9.76 | 12.73 |
> |            |               | StyleGAN2 |       |       |   9.62 | 12.74 |       |       |   9.64 | 12.76 |
> | StyleGAN2  | CelebA30k     | MCL-GAN   | 40.18 |  9.18 |  39.89 |  9.26 | 40.14 |  9.21 |  39.84 |  9.27 |
> |            |               | StyleGAN2 |       |       |  39.72 |  9.53 |       |       |  39.79 |  9.54 |

---

> > ### Author Response · Authors · 2022-08-02
> > **Response #2**
> >
> > 2. We additionally measured precision, recall, density, and coverage using [2] as suggested by the reviewer. The two tables below correspond to Table 1 and 4 of the main paper, respectively. We discover that the general trend is similar to the old version of precision/recall while we consistently observe the improvement in recall performance (diversity). In particular, Inclusive GAN records significantly poor recall scores but high scores for all other metrics. This implies the model tends to memorize the real dataset and fails to generate diverse unseen samples.
> >
> > \
> > **[DCGAN backbone]**
> > |         |       |    |   MNIST   |        |         |          | Fashion |MNIST|         |          |   CelebA  |        |         |          |
> > |---------|-------|----|:---------:|:------:|:-------:|:--------:|:-------------:|:------:|:-------:|:--------:|:---------:|:------:|:-------:|:--------:|
> > | **Method** | **loss**| **M** |**prec.**|**rec.**|**den.**|**cvg.**|**prec.**|**rec.**|**den.**|**cvg.**|**prec.**|**rec.**|**den.**|**cvg.**|
> > | Base    |  GAN |  1 |   0.121   |  0.549 |  0.034  |   0.094  |     0.576     |  0.190 |  0.363  |   0.324  |   0.509   |  0.304 |  0.285  |   0.395  |
> > | GMAN    |       |  5 |   0.445   |  0.548 |  0.187  |   0.384  |     0.734     |  0.551 |  0.766  |   0.684  |   0.602   |  0.216 |  0.468  |   0.479  |
> > | GMAN    |    | 10 |   0.491   |  0.555 |  0.215  |   0.421  |     0.751     |  0.540 |  0.834  |   0.760  |   0.646   |  0.294 |  0.626  |   0.640  |
> > | Ours |       |  5 |   0.564   |  0.792 |  0.309  |   0.562  |     0.795     |  0.676 |  0.823  |   0.819  |   0.681   |  0.367 |  0.647  |   0.672  |
> > | Ours |       | 10 |   0.525   |  0.776 |  0.274  |   0.511  |     0.777     |  0.666 |  0.817  |   0.798  |   0.619   |  0.418 |  0.454  |   0.567  |
> > | Base    |LSGAN |  1 |   0.362   |  0.697 |  0.146  |   0.334  |     0.472     |  0.378 |  0.261  |   0.326  |   0.688   |  0.322 |  0.616  |   0.620  |
> > | GMAN    |  | 10 |   0.465   |  0.513 |  0.205  |   0.398  |     0.754     |  0.552 |  0.875  |   0.759  |   0.616   |  0.312 |  0.546  |   0.622  |
> > | Ours |       | 10 |   0.586   |  0.794 |  0.379  |   0.598  |     0.772     |  0.666 |  0.723  |   0.771  |   0.678   |  0.372 |  0.675  |   0.699  |
> > | Base    | Hinge |  1 |   0.103   |  0.288 |  0.029  |   0.076  |     0.509     |  0.281 |  0.304  |   0.347  |   0.559   |  0.353 |  0.382  |   0.512  |
> > | Ours |  |  5 |   0.416   |  0.644 |  0.175  |   0.352  |     0.559     |  0.373 |  0.371  |   0.418  |   0.625   |  0.347 |  0.521  |   0.583  |
> > | Ours |       | 10 |   0.506   |  0.741 |  0.251  |   0.472  |     0.572     |  0.350 |  0.380  |   0.439  |   0.583   |  0.358 |  0.414  |   0.548  |
> >
> > **[StyleGAN2 backbone]**
> > |               |  CIFAR-10  |        |         |          | CelebA30k |        |         |          |
> > |---------------|:---------:|:------:|:-------:|:--------:|:---------:|:------:|:-------:|:--------:|
> > | **Method**      |**prec.**|**rec.**|**den.**|**cvg.**| **prec.**|**rec.**|**den.**|**cvg.**|
> > | StyleGAN2     |   0.781   |  0.532 |  1.139  |   0.853  |   0.747   |  0.385 |  0.905  |   0.835  |
> > | Inclusive GAN |     -     |    -   |    -    |     -    |   0.916   |  0.099 |  2.000  |   0.763  |
> > | Ours       |   0.786   |  0.553 |  1.173  |   0.861  |   0.803   |  0.414 |  1.105  |   0.880  |
> >
> > \
> > 3. Regarding the comments on identifying minority groups, the additional experiments on Fashion-MNIST and partial MNIST show how well the MCL-GAN captures the minority mode using multiple discriminators. Please refer to C.1 of our response to reviewer FAZc.
> >
> > \
> > [1] A. K. Mondal et al., FlexAE: Flexibly Learning Latent Priors for Wasserstein Auto-Encoders, UAI 2021. \
> > [2] M. F. Naeem et al., Fidelity and Diversity Metrics for Generative Models, ICML 2020.

---

### Official Review · Reviewer_bp3P · 2022-07-09

**Rating:** 5
**Confidence:** 5
**Soundness:** 3 good
**Presentation:** 3 good
**Contribution:** 1 poor

**Summary:**

This paper proposes a new GAN framework named MCL-GAN. MCL-GAN comprises a single generator and multiple discriminators and follows a multiple-choice learning (MCL) strategy to forward signals of samples to the generator. By conveying the plentiful signals to the generator, the authors insist the generator can generate diverse samples, which means MCL is effective in overcoming the chronic mode-collapse problem. By performing image generation experiments using standard benchmark datasets, the authors demonstrate that the proposed MCL-GAN helps in increasing the diversity of generated images.

**Questions:**

No

**Ethics Review Area:**

["I don’t know"]

**Limitations:**

I understand that the authors are cautious about discussing the shortcomings of their work. So, I think it does not matter whether the authors provide the limitations of MCL-GAN or not. Instead, I highly recommend the authors write more about the negative societal impact of their work in the supplementary material.

**Strengths And Weaknesses:**

Strengths

(+) The motivation of MCL-GAN is reasonable. Also, MCL-GAN is simple, so I think researchers can implement MCL-GAN easily.

(+) MCL-GAN does not require too many parameters to be burdensome. Multiple discriminators share their parameters, and all we need is stacking multiple fully connected layers parallelly upon the shared network.

(+) Toy experiments are appropriately designed and successfully support the main pillar of the paper.

(+) Experiments using benchmark datasets demonstrate that the proposed MCL training is effective, and MCL-GAN can generate more diverse images than a baseline.

Weaknesses

(-) Limited novelty. Although experimental results show the efficacy of MCL-GAN, I think the technical novelty of MCL-GAN is limited. I think introducing MCL into some frameworks is not new to the community.

(-) I am not sure whether the proposed MCL-GAN can be scaled up for high-resolution image generation. This paper does not contain any experiment using high-resolution image datasets, such as ImageNet, AFHQ, and FFHQ.

(-) Using the old version of precision and recall [1] seems improper for measuring the diversity of generated images. I recommend the authors use the improved version of precision and recall [2] to quantify the diversity. The paper [2] has shown that the old precision and recall sometimes cannot capture the diversity well. The authors can compare their model with previous methods using StudioGAN library [3].

[1] Sajjadi, M.S., Bachem, O., Lucic, M., Bousquet, O., & Gelly, S. (2018). Assessing Generative Models via Precision and Recall. NeurIPS.

[2] Kynkäänniemi, T., Karras, T., Laine, S., Lehtinen, J., & Aila, T. (2019). Improved Precision and Recall Metric for Assessing Generative Models. ArXiv, abs/1904.06991.

[3] https://github.com/POSTECH-CVLab/PyTorch-StudioGAN.

---

> ### Author Response · Authors · 2022-08-02
> **Response**
>
> We appreciate the time taken to provide us with valuable feedback. We respond to each of the concerns and suggestions below.
>
>
> 1. The proposed approach is the first attempt to combine multiple choice learning (MCL) and generative adversarial network (GAN), which involves non-trivial challenges. In particular, the design of the loss function is not straightforward because it has to solve a minimax problem with separate loss terms for the generator and the discriminators while considering two different cases, for expert and non-expert training. In addition, we introduce the balance term for the stability of training, and optionally adopt the L1 regularization term to control the effective number of discriminators. Therefore, we believe that our contribution is beyond the simple integration of two existing techniques and has sufficient contribution. Note that, since MCL-GAN is based on a single generator, it does not incur an additional computational cost for inference and can be easily combined with arbitrary GAN models.
>
>
> 2. We primarily focus on the proof of our concept on several GAN losses based on the basic architecture, i.e., DCGAN, which would be more helpful to understand the behavior of the proposed method and then extend the application to more complex and large models with various tasks. Due to the limited resources and time available, we will do our best to report additional results on larger resolutions during the discussion period. However, we ask reviewers to read our supplementary document, which contains a lot of experiments on various tasks. We believe that it can be helpful to better understand the benefits of MCL-GAN.
>
>
> 3. We additionally computed the more recent version of precision/recall[1] along with density/coverage proposed in [2]. Please refer to our response #2 to the reviewer Hs6x.
>
>
> [1] T. Kynkäänniemi, et al., Improved Precision and Recall Metric for Assessing Generative Models. NeurIPS 2019.
>
> [2] M. F. Naeem et al., Fidelity and Diversity Metrics for Generative Models, ICML 2020.

---

> > ### Author Response · Authors · 2022-08-04
> > **Limitations**
> >
> > We describe a more detailed discussion about the limitations of our work below.
> >
> > The proposed method carries additional hyperparameters including the weights for several loss terms and the number of discriminators, and one might question the robustness of MCL-GAN with respect to the variations of the hyperparameters. From our analysis of the hyperparameter setting, the performance of the proposed method improves significantly by the expert training and the balanced assignment of discriminators while the rest of the loss terms make stable contributions over a wide range of their weights. Also, since MCL-GAN adjusts the number of active discriminators that participate in learning as experts, its performance is robust to the number of discriminators. Although not included in the scope of this paper, there are some settings that need further investigation. For example, how to extend the multi-discriminator environment with an extremely small dataset and compatibility with recent data augmentation techniques are not discovered but are worth studying.

---

> > > ### Comment · Reviewer_bp3P · 2022-08-07
> > > **Retaining my score**
> > >
> > > 1. As Reviewer oqkx stated, I still think that the idea of MCL-GAN is incremental. Using multiple discriminators [1, 2, 3] and sharing discriminator's parameters [4] are actively adopted in the GAN community. Also, there is no theoretical explanation for why MCL-GAN can generate more diverse images compared to the baseline. For these reasons, I think the technical novelty of MCL-GAN is limited.
> > >
> > > 2. I don't think it's enough to experiment with small data and simple architectures. Since the results are different when experimenting with small and large data, I am not sure whether the proposed method really works.
> > >
> > > 3. Thank you for providing more experiments using the improved precision and recall, and the negative societal impact of authors' work.

---

> > > > ### Author Response · Authors · 2022-08-08
> > > > **Experiment on the large resolution**
> > > >
> > > > Thank you for your patience. We ran a StyleGAN2 experiment on 70k 512×512 FFHQ data using the Config E architecture and report the FID alongside the precision/recall/density/coverage[1,2] below. While precision is slightly decreased for MCL-GAN, the recall score is enhanced with a much larger gap, and the overall performance is improved as shown by the FID and F1 scores. The comparison of density is rather meaningless as both scores are greater than 1. Note that the number of parameters increases by only 0.0071% which is insignificant compared to the large capacity of the base architecture. We hope that these results have cleared any doubts regarding the effectiveness of our method on large resolutions.
> > > >
> > > > |               |  FID | precision | recall | density | coverage  | F_1 score |
> > > > |---------------|:----:|:---------:|:------:|:-------:|:---------:|:---------:|
> > > > | StyleGAN2     | 3.87 |   0.801   |  0.590 |  1.260  |   0.949   |   0.679   |
> > > > | MCL-GAN (M=5) | 3.53 |   0.794   |  0.636 |  1.178  |   0.952   |   0.706   |
> > > >
> > > > We would also like to emphasize that the proposed method does not alter any architecture/capacity of the generator or generation process. Although this may ​​impose a limit on the amount of performance improvement, we believe the method is meaningful because it can be applied to a wide variety of frameworks, regardless of the model architecture and task setting.

---

### Official Review · Reviewer_oqkx · 2022-07-10

**Rating:** 4
**Confidence:** 4
**Soundness:** 3 good
**Presentation:** 4 excellent
**Contribution:** 3 good

**Summary:**

This paper proposes a generative adversarial network with multiple discriminators for GAN training. The proposed method guides each discriminator to have expertise in the subset of the entire data. For each sample, only k experts out of M discriminators are used. To address the over-confident problem, this paper proposes another loss, named non-expert training loss. To balance the assignment of discriminators, this paper proposes another loss, named balance loss. The proposed method is evaluated on several tasks, including unconditional image generation, image-to-image translation, and text-to-image synthesis. The results show that the proposed method outperforms the baselines.

**Questions:**

Please refer to Weaknesses.

**Ethics Review Area:**

["I don’t know"]

**Limitations:**

In practice, it may be difficult to tune the non-expert training loss and balance loss.

**Strengths And Weaknesses:**

Strengths:
1. The idea of using multiple discriminators is sensible.
2. The proposed method is evaluated on various tasks, including unconditional image generation, image-to-image translation, and text-to-image synthesis.
3. The experiments show that the proposed method outperforms GMAN.

Weaknesses:
1. The proposed method is an extension of GMAN. The difference is that the proposed method selects part of the multiple discriminators. In my opinion, this extension is incremental.
2. The extension (i.e., using part of the multiple discriminators) also introduces two problems, i.e. over-confident problem and balance problem. Although the authors propose two specific losses to tackle the above problems, both the over-confident problem and balance problem seem difficult to tune in practice.
3. The advantages of the proposed method over GMAN are not clearly explained. This paper stated that “While GMAN focuses on the loss to the generator with parallel learning of discriminators, our strategy specializes each discriminator for more informative feedbacks to the generator”. This is not convincing to me.
4. The resolutions of images used in the experiments are 32x32, 64x64, or 128x128. I think the effectiveness of the proposed method should be verified on larger datasets, such as FFHQ with 1024x1024.

---

> ### Author Response · Authors · 2022-08-02
> **Response**
>
> We appreciate the time taken to provide us with valuable feedback. We respond to each of the concerns and suggestions below.
>
>
> 1. The main difference between our method and GMAN is the specialization of discriminators. We use the max-criterion to train the generator but also to train the discriminators to be specialized on the subset of the real dataset. While the discriminators of GMAN do not share the parameters, our method shares the model parameters except for the last layer, which in turn, makes our algorithm more computationally efficient and free from scalability issues for a model with a large discriminator (see Section D of the supplementary file). Besides, GMAN reports that max-critic does not work well in practice and better ensemble results are obtained from the soft-critic. We also discover that GMAN fails to converge with some widely-used losses, e.g., Hinge loss.
>
>
> 2. The performance is not critically sensitive to the coefficients of the balance losses and non-expert training and surpasses a single discriminator GAN with a large margin throughout the wide range of the choice for the coefficient. We provide the ablation studies on hyperparameters in Section C of the supplementary file.
>
>
> 3. In terms of training discriminators, while discriminators of GMAN are independently trained on the whole dataset, MCL-GAN causes the inter-connection of discriminators by their specialization on the subset of data. Unlike GMAN, which differentiates discriminators by hyperparameters such as architectural design and training initialization, etc, our method naturally leads to the specialization of discriminators on semantic subgroups by the simple strategy and design of losses.
>
>
> 4. We primarily focus on the proof of our concept on several GAN losses based on the basic architecture, i.e., DCGAN, which would be more helpful to understand the behavior of the proposed method, and then extend the application to more complex and large models with various tasks. For conditioned image synthesis experiments (Sec 5.3), we used 256x256-sized images. Due to the limited resources and time available, we will do our best to report additional results on larger resolutions during the discussion period. However, we ask reviewers to read our supplementary document, which contains a lot of experiments on various tasks. We believe that it can be helpful to better understand the benefits of MCL-GAN.

---

> > ### Author Response · Authors · 2022-08-08
> > **Experiment on the large resolution**
> >
> > Thank you for your patience. We ran a StyleGAN2 experiment on 70k 512×512 FFHQ data using the Config E architecture and report the FID alongside the precision/recall/density/coverage[1,2] below. While precision is slightly decreased for MCL-GAN, the recall score is enhanced with a much larger gap, and the overall performance is improved as shown by the FID and F1 scores. The comparison of density is rather meaningless as both scores are greater than 1. Note that the number of parameters increases by only 0.0071% which is insignificant compared to the large capacity of the base architecture.
> >
> > |               |  FID | precision | recall | density | coverage  | F_1 score |
> > |---------------|:----:|:---------:|:------:|:-------:|:---------:|:---------:|
> > | StyleGAN2     | 3.87 |   0.801   |  0.590 |  1.260  |   0.949   |   0.679   |
> > | MCL-GAN (M=5) | 3.53 |   0.794   |  0.636 |  1.178  |   0.952   |   0.706   |
> >
> > [1] T. Kynkäänniemi, et al., Improved Precision and Recall Metric for Assessing Generative Models. NeurIPS 2019.
> >
> > [2] M. F. Naeem et al., Fidelity and Diversity Metrics for Generative Models, ICML 2020.

---

> > ### Comment · Reviewer_oqkx · 2022-08-10
> > **Thanks for the response**
> >
> > Thanks for the response. I still have the concern that this extension to GMAN is incremental. Thus, I would like to keep my original rating.

---

### Official Review · Reviewer_FAZc · 2022-07-12

**Rating:** 4
**Confidence:** 4
**Soundness:** 4 excellent
**Presentation:** 3 good
**Contribution:** 2 fair

**Summary:**

The paper proposes a simple yet effective way to mitigate the mode collapse issue of GANs. To this end, the authors propose the utilization of multiple discriminators that specialize on different subsets of the distribution. This process is implemented using multiple choice learning.

**Questions:**

I suggest that the authors address the issues raised in "weaknesses" section.

**Limitations:**

The authors have included an ethical considerations part, however I would also like to see a discussion regarding the limitations of the proposed method.

**Strengths And Weaknesses:**

Strengths:

 A. The main idea is simple and clearly presented.

 B. The experimental results highlight the efficacy of the method in the terms of a number of metrics.


Weaknesses:

A. In order to understand the significance of the proposed method, I’m missing a comparison with other clustering-based approaches, e.g.,

[1]. ClusterGAN : Latent Space Clustering in Generative Adversarial Networks

[2]. Diverse Image Generation via Self-Conditioned GANs


B.  I would like to see some more metrics that are specific to the diversity of the generated samples. e.g., IvOM [3],  Pairwise Distance [3].

[3]. Unrolled generative adversarial networks


C. I would like to see how the approach performs on some mode-coverage benchmarks, e.g.,

 - Fashion-MNIST and partial MNIST of [4]

 - Stacked MNIST of  [5]

[4]. Rethinking Generative Mode Coverage: A Pointwise Guaranteed Approach

[5]. VEEGAN: Reducing Mode Collapse in GANs using Implicit Variational Learning

---

> ### Author Response · Authors · 2022-08-02
> **Response**
>
> We appreciate the time taken to provide us with valuable feedback. We respond to each of the suggestions below.
>
>
> **A. Comparison to clustering-based approaches**
>
> We compare the results on CIFAR-10 with two clustering-based approaches[1,2] that the reviewer mentioned. We run the experiment on two different network architectures---Type 1) DCGAN used for results in Table 2 of the main paper and Type 2) CNN used in [2]. We present the results along with some results in Table 2 of our main paper. Note that the asterisk (*) denotes the copied results from its original paper. In both backbones, MCL-GAN achieves better FID scores than clustering-based methods. This is promising because [2] relies on a clustering algorithm which causes additional computation and has more model parameters for the generator (~x1.5) due to cluster conditioning on the generator’s input.
>
> | Method                  |       FID       | Remark       |
> |-------------------------|:---------------:|--------------|
> | **Type 1 backbone**|                 |              |
> | DCGAN                   |       37.7      |              |
> | GMAN                    |      37.11      | 10 Discs     |
> | [5]                     |      30.26      | 10 Discs     |
> | ClusterGAN[1]           |      40.36      | 10 clusters  |
> | Self-Cond. GAN[2] |      31.44      | 100 clusters |
> | MCL-GAN                 |      26.87      | 10 Discs     |
> | **Type 2 backbone**|                 |              |
> | Self-Cond. GAN[2]*| 18.70 ± 1.28 | 100 clusters |
> | MCL-GAN                 | 17.15 ± 0.12 | 5 Discs      |
>
>
> **B. Pairwise Distance**
>
> We additionally measured L2 Pairwise Distance as proposed in [3] for 4 datasets. We visualize the histograms of pairwise distances of samples for each model. Please see Figure A of the “figures.pdf” file of the supplementary material. The vertical lines of each plot indicate the median values of the corresponding statistics and we also provide approximate KL divergences with respect to real data in the legends of the plots. In most cases, MCL-GANs have the closest distribution of the pairwise distance to those of the real dataset and the median values are not significantly smaller than those of real data unlike some cases for GMAN and Inclusive GAN.
>
>
> **C. Mode-coverage benchmarks**
>
>
> ***C.1. Fashion-MNIST and partial MNIST***
>
> For this benchmark, we follow the same settings of [4] which trains with 60k Fashion-MNIST and 100 1’s of MNIST. We show our results below. The results for other methods are copied from [4]. We discover that MCL-GAN successfully captures the minority and its frequency is very close to the ideal one. Furthermore, the quality of randomly sampled 1’s is higher than those with the highest prediction scores of other methods. (Please see Figure B in the “figure.pdf” in the supplementary file.)
>
> | Method  |         |  # of 1's   | Frequency | Avg Prob. |
> |---------|---------|:---------:|:---------:|:---------:|
> | DCGAN   |         |     13    |  0.00014  |    0.49   |
> | MGAN    | 30 Gen  | collapsed |     -     |     -     |
> | AdaGAN  | 30 Gen  |     60    |  0.00067  |    0.45   |
> | [4]     | 30 Gen  |    289    |  0.00321  |    0.68   |
> | MCL-GAN | 10 Disc |    153    |  0.00170  |    0.84   |
> | Ideal   |         |   149.8   |  0.00166  |    1.00   |
>
> ***C.2. Stacked MNIST***
>
> For this benchmark, we use 240k 32x32-sized stacked MNIST data for training and evaluated the model on randomly generated 26k samples. we compare the results with DCGAN and GMAN on the same backbone. Note that we found it difficult to achieve reasonable quality when training GMAN with 10 discriminators on this dataset.
>
> |                |    Mode count   | KL (model \| data) |
> |----------------|:---------------:|:------------------:|
> | DCGAN          |       848       |        0.51        |
> | GMAN (M=10)    | 143  |        1.54        |
> | MCL-GAN (M=10) |       994       |        0.34        |
>
>
> [1] S. Mukherjee et al., ClusterGAN: Latent Space Clustering in Generative Adversarial Networks, AAAI 2019.
>
> [2] S. Liu et al., Diverse Image Generation via Self-Conditioned GANs, CVPR 2020.
>
> [3] L. Metz et al., Unrolled generative adversarial networks, ICLR 2017.
>
> [4] P. Zhong et al., Rethinking Generative Mode Coverage: A Pointwise Guaranteed Approach, NeurIPS 2019.
>
> [5] I. Albuquerque, et al., Multi-objective training of generative adversarial networks with multiple discriminators, ICML 2019.

---

> > ### Author Response · Authors · 2022-08-04
> > **Limitations**
> >
> > We describe a more detailed discussion about the limitations of our work below.
> >
> > The proposed method carries additional hyperparameters including the weights for several loss terms and the number of discriminators, and one might question the robustness of MCL-GAN with respect to the variations of the hyperparameters. From our analysis of the hyperparameter setting, the performance of the proposed method improves significantly by the expert training and the balanced assignment of discriminators while the rest of the loss terms make stable contributions over a wide range of their weights. Also, since MCL-GAN adjusts the number of active discriminators that participate in learning as experts, its performance is robust to the number of discriminators.
> > Although not included in the scope of this paper, there are some settings that need further investigation. For example, how to extend the multi-discriminator environment with an extremely small dataset and compatibility with recent data augmentation techniques are not discovered but are worth studying.

---

### Author Response · Authors · 2022-08-02
**General response to reviewers**

We appreciate all the reviewers for their time spent providing us with valuable feedback. We responded to each of the comments individually and attached a separate additional file that contains the visual data regarding our responses in the supplementary material. Also, we would like the reviewers to take into account the supplementary file as we provide more analysis with extensive experiments on behaviors of MCL-GAN. We believe that more visualizations and analyses related to discriminator specializations and hyperparameters analysis can be helpful to better understand the benefits of MCL-GAN.

---

### Author Response · Authors · 2022-08-08
**Experiment on the large resolution & Request for further discussions**

We conducted an additional experiment with StyleGAN2 on the FFHQ dataset of 512x512 resolution, and posted our results in the responses to reviewers `oqkx` and `bp3P`. Please let us know if you have any further issues or questions. We look forward to further discussions to clarify any other concerns.

---

### Author Response · Authors · 2022-08-10
**Summary of rebuttal**

We summarize the additional experiments and clarifications on the concerns regarding the issues raised by reviewers. We have responded to all concerns regarding experimental results and further evaluations raised by the reviewers. We also request the reviewers to refer to the supplementary file, since we have provided additional visualizations to clear out doubts.


1. At the suggestion of reviewer `FAZc`, we compared the proposed method with two other clustering-based methods.

2. We evaluated the diversity of the samples generated by our method using an extra metric (pairwise distance), and compared the capability of mode coverage using two benchmarks. These experimental results showed superiority over other compared methods.

3. For better quantification on precision/recall, as requested by reviewers `bp3P` and `Hs6x`, we re-evaluated the main results of our original submission using the recent version of precision/recall as well as the density/coverage and showed the consistency of our results.

4. We conducted a StyleGAN2 experiment on the 512x512-resolution FFHQ dataset to validate the efficacy of the proposed method on larger resolutions to address the concerns of reviewers `oqkx` and `bp3P`.

5. Regarding the concern whether the method is overfitting on the training set raised by reviewer `Hs6x`, we computed additional measurements on Pixel/Inception Memorization Scores and added the visualization of nearest neighbors from training images.


We hope that our additional experiments and evaluations have addressed the concerns raised by reviewers and cleared some doubts regarding the proposed method. Thank you.

---

### Meta-Review · Area_Chair_DchV · 2022-08-26

**Recommendation:** Accept
**Confidence:** Less certain

**Metareview:**

The paper considers a GAN setting with multiple discriminators, a topic that has been studied quite a few times before. It proposes a novel approach based on multiple choice learning in order to specialize discriminators to certain modes of the data distribution, with the aim of mitigating collapse. They demonstrate this clustering effect qualitatively, and present quantitative results on a number of tasks. Parameter-sharing keeps the computational burden in check, while the number of discriminators is determined adaptively.

Reviewers generally found the problem of interest, and the method simple, reasonably motivated, not burdensome in the additional hyperparameters, and generally applicable. A common complaint was the lack of high resolution results. While it is important not to unduly limit acceptance of papers wherein the authors are operating under a constrained compute budget, I believe that at this stage the request for some megapixel was reasonable, and of scientific interest as techniques which show promise at smaller scales often show little at larger scales.

The authors did provide these results in rebuttal, along with a great number of other requested comparisons and revisions that I feel improve the empirical rigour of the work. As of the end of the discussion period, two reviewers (oqkx, FAZc) lean reject; FAZc did not respond to, or acknowledge, the rebuttal, in which it appears to me their concerns were addressed, and therefore I am discounting their score in my evaluation. oqkx's concerns can be summarized as a) incrementality; b) speculation on tuning difficulty; c) unconvinced by the explanation of advantages over GMAN; d) scale of experiments. The rebuttal seems to have addressed all of oqkx's concerns except a).

I don't believe that concerns around incrementality of the approach justify rejection in this instance. The method is quite clearly differently motivated from GMAN, the appeal to multiple choice learning is to my knowledge novel and appears sensible, the empirical results (which the authors have gone the extra mile to improve during the review period) are extensive and convincing. Despite the borderline scores, I tentatively recommend acceptance.

**Award:**

No

---

### Decision · Program_Chairs · 2022-09-14

Accept